# Alternative Methods of Ant (Hymenoptera: Formicidae) Control with Emphasis on the Argentine Ant, *Linepithema humile*

**DOI:** 10.3390/insects12060487

**Published:** 2021-05-24

**Authors:** Daniel R. Suiter, Benjamin M. Gochnour, Jacob B. Holloway, Karen M. Vail

**Affiliations:** 1Department of Entomology, University of Georgia Griffin Campus, 1109 Experiment Street, Griffin, GA 30223, USA; 2D.B. Warnell School of Forestry and Natural Resources, University of Georgia, Athens, GA 30602, USA; benjamin.gochnour@uga.edu; 3U.S. Army Environmental Command, Fort Sam Houston, San Antonio, TX 78234, USA; jbh301@gmail.com; 4Department of Entomology, University of Tennessee, Knoxville, TN 37996, USA; kvail@utk.edu

**Keywords:** Argentine ant, *Linepithema humile*, essential oil, trail pheromone, hydrogel, ant trapping, baits, alternative ant control, fipronil, (Z)-9-hexadecenal

## Abstract

**Simple Summary:**

Ants can be major pests to homeowners and other property owners. In the U.S., ants often rank as one of the most common and difficult-to-control pests around and in homes, businesses, and other facilities. Typically, ant control practices are conducted by licensed pest management professionals with sprays, baits, and granular products, containing various types of insecticides, applied to the outside perimeter of infested structures. Many of the insecticides used to control pest ants are harmful to non-target organisms, especially those in aquatic environments. To address these negative environmental impacts, research on alternative, generally low-impact and least toxic, ant control strategies has received a fair amount of attention. The underlying goal of this research is a reduction in human exposure to traditional insecticides. Examples of alternative approaches included in this review article include the use of essential oils and other chemicals as deterrents to ant nesting and foraging; ant trail pheromones as disruptants to foraging; mass trapping of ants; new gel baits containing extremely low concentrations of insecticide; and altering the behavior and distribution of ants by altering their access to food.

**Abstract:**

Ants (Hymenoptera: Formicidae), especially the Argentine ant, *Linepithema humile* (Mayr), can be significant nuisance pests in urban and suburban environments. Conventional interventions have primarily relied on the use of chemical insecticides, namely fipronil and bifenthrin, applied as residual, contact treatments around the outside perimeter of infested structures. Despite tightening regulation limiting the scope of insecticide applications in urban settings, dependence on these products to manage ants continues, resulting in significant water contamination. The U.S. EPA, in response, has further restricted the use patterns of many insecticides used for ant control in professional and over-the-counter markets. The purpose of this review is to summarize the relevant literature associated with controlling nuisance pest ants, with emphasis on *L. humile*, without the use of liquid broadcast applications of EPA-registered insecticides while focusing on low-impact, alternative (to broadcast applications) pest control methods. Specific subsections include Trail Pheromone; Use of Behavior-Modifying Chemicals; Mass Trapping; Hydrogels, “Virtual” Baiting, and Exceedingly-Low Bait Concentrations; Food Source Reduction; Deterrents; and RNA Interference (RNAi).

## 1. Pest Ant Control around Homes

Surveys of the U.S. structural pest control industry routinely rank ants as one of the most common nuisance pests around homes. Like other invasive ant species, the Argentine ant, *Linepithema humile* (Mayr), is an ecologically disruptive and economically important pest. While an economic pest in agricultural settings, *L. humile* is a major nuisance pest in urban and suburban environments, where it is often found in large numbers foraging in and around homes and other buildings [1]. Historically, ant control around homes has been conducted by the structural pest control industry using liquid, granular, and gel baits; liquid, residual contact insecticides (sprays); and granular formulations applied to soil, mulch, vegetation, perimeter walls, and potential points of entry of infested structures [2]. Klotz et al. [3,4,5,6,7] evaluated, for Argentine ant control around residences, residual, contact sprays containing bifenthrin, chlorpyrifos, cyfluthrin, deltamethrin, lambda-cyhalothrin, and fipronil; a granular product containing bifenthrin; and baits containing hydramethylnon, disodium octoborate tetrahydrate, fipronil, thiamethoxam, imidacloprid, and metaflumizone. All the products and strategies provided some relief from nuisance ant infestations, but because of the supercolony nature of Argentine ants, the complexity of the substrates treated, and the sometimes-harsh environmental conditions during and after treatment, many products and strategies began to lose effectiveness after six to eight weeks. Liquid sprays containing fipronil were consistently the most effective and longest-lasting treatment, and this product has been widely adopted for nuisance ant control since its introduction into the ant control market in the U.S. in 2002. Fipronil spot treatments to trailing ants (3.8 L per home) and 5 cm wide pin stream treatments (3.8 L applied to the home’s perimeter base) were as effective as a traditional fipronil broadcast spray (15.2 L per home), resulting in a 75% reduction in applied fipronil with no sacrifice in ant control, i.e., ≈80% to 90% reduction in ant feeding activity 6 to 8 weeks after treatment [4,6,7]. Given the ease and effectiveness of fipronil spray applications, baiting for Argentine ants is less common among practitioners. Baiting too is slightly less effective at controlling Argentine ants and is certainly more labor intensive and costly from the practitioner’s point of view. However, a 0.001% thiamethoxam bait provided Argentine ant control comparable to a liquid fipronil spray [6].

## 2. Standard Ant Control Methods Can Be Problematic

The use of residual contact chemical insecticides to control nuisance infestations of ants raises concern about the realized and potential drawbacks to their use. These concerns include secondary ecological effects, such as non-target and un-intended, negative environmental impacts, especially in aquatic habitats; improper use; insecticide run-off and water contamination, primarily driven by irrigation systems and application close to watershed sites; and costs. Although *L. humile* does not pose a direct health threat to humans, the continued reliance on chemical inputs to control them poses significant environmental risks. These concerns have led to significant regulatory changes in recent years that have limited the type and use of chemical measures for ant control. In 2013, the U.S. Environmental Protection Agency (EPA), in an effort “to reduce ecological exposure from residential uses of pyrethroid and pyrethrin products,” implemented an initiative to revise guidelines for pyrethroid- and pyrethrin-based pesticide products used in non-agricultural outdoor settings [8]. This initiative specifically limited non-agricultural outdoor uses of pyrethroids and pyrethrins to spot or crack-and-crevice treatments only with a few exceptions. Justifications for the implementation of these changes are well documented and were recently reviewed [9]. For example, Greenberg et al. [10] evaluated fipronil and bifenthrin levels in run-off and irrigation water directly following residential home treatments for Argentine ants. The 2007 treatment consisted of a spray application of the two insecticides using standard practices. The 2008 treatment applied product as a pin stream, utilized spray-free zones, and restricted insecticide applications to the foundation of the homes. After one week, the fipronil level in the run-off water from the 2007 treatment was enough to cause acute toxicity to sensitive, aquatic organisms. At eight weeks post-treatment, the same was true for bifenthrin levels in the run-off water. In contrast, the water samples from the 2008 treatment contained no detectable fipronil at one week post-treatment and contained greatly reduced levels of bifenthrin compared to the 2007 treatment. Jiang et al. [11] evaluated fipronil toxicity to Argentine ants and run-off potential on concrete surfaces after exposure to summer weather conditions and simulated precipitation. Fipronil-treated surfaces killed >50% of Argentine ants, exposed for one minute, within 16 h of exposure. Toxicity was lost on these surfaces after 20 days while run-off water from simulated precipitations contained detectable amounts of insecticide 89 days after treatment. Gan et al. [12] collected run-off water from large, residential communities, each comprised of hundreds of single-family homes in Orange County and Sacramento County, California. The water samples were tested for fipronil and its biologically active derivatives over 26 months. The amount of fipronil in the samples from Orange County was greater than 10-fold the amount of that in Sacramento County. In Orange County, fipronil load in water was positively correlated with higher insecticide use. A temporal pattern was also observed, with higher fipronil levels in the samples from April to October, while levels decreased from October through March. This pattern reflects the higher activity of Argentine ants in the warmer months resulting in greater use of insecticides and coincides with more frequent rain events leading to more insecticide being washed away in the run-off water. Fipronil and its oxidative derivative, fipronil sulfone, comprised 70% of the total concentration of insecticides in the samples, while fipronil’s photolytic derivative, desulfinyl fipronil, accounted for another 25% of the concentration. The fipronil and fipronil derivative levels found in the water samples often exceeded the LC50 values for local, aquatic arthropods.

## 3. Alternative Methods of Ant Control

In light of environmental concerns noted above, alternative ant control methods, defined as having low to no impact on non-target organisms, have received increasing attention with particular interest in the use of “natural” products with reported repellent and/or insecticidal activity, including essential oils as repellents and insecticides; hydrogels; and pheromones, alone and in conjunction with other products. Reviews of alternative pest management techniques for Argentine ants were published previously [2,13].

### 3.1. Trail Pheromone

In a series of studies, the usefulness of (Z)-9-hexadecenal, historically perceived and reported to be a component of Argentine ant trail pheromone, as a disruptor of ant foraging was explored [14,15,16,17,18,19]. Contrary to long-held beliefs, (Z)-9-hexadecenal, although present in Argentine ants in minimal quantities, was not detectable as a component of the Argentine ant trail pheromone [20]. The trail pheromone components were dolichodial and iridomyrmecin. Interestingly, at unnaturally high concentrations, synthetic (Z)-9-hexadecenal was highly attractive to Argentine ants in olfactometer bioassays. Regardless, (Z)-9-hexadecenal might be useful as a method to control Argentine ants by complicating their ability to find food and recruit. Ants co-occurring with Argentine ants became more competitive for food resources when (Z)-9-hexadecenal saturated the environment [21]. When delivered from a point source near trailing ants in the field (Z)-9-hexadecenal disrupted normal trailing behavior and caused ants to run about chaotically and to abandon the natural pheromone trail laid down by foragers [14,15]. More importantly, when large field plots were treated with the pheromone, ant foraging was disrupted to the point that they were less efficient at finding and recruiting to new and existing food resources. The technique is similar to mating disruption, where sex pheromone saturation of a target environment results in the male’s inability to locate receptive females. In the case of (Z)-9-hexadecenal ants lose the ability to locate and recruit to food. A microencapsulated formulation of the pheromone produced similar results and extended the product’s residual life to 14 days under field conditions [16]. In 100 square meter plots in Japan, saturation with monthly releases of (Z)-9-hexadecenal suppressed Argentine ant foraging by 70% over an entire season of ant activity (April to November) [17]. Unfortunately, the number of foraging ants in plots was not reduced probably because of migration from untreated into treated plots. In a follow-up study, Argentine ant numbers were reduced only when a toxic bait was provided to the ants in the presence of (Z)-9-hexadecenal release [19]. In the absence of chemical treatments, Nishisue et al. [17] propose that the utility of disruption of foraging Argentine ants should be areawide, instead of local, to allow for the large colony size (supercolonies) of this invasive ant.

There are other uses of (*Z*)-9-hexadecenal. Greenberg and Klotz showed that adding (Z)-9-hexadecenal to a sucrose solution enhanced feeding by Argentine ants in laboratory and field tests [22]. In the lab experiment, the pheromone enhanced feeding > 150%. In the field, the pheromone was applied to a plastic membrane covering the end of a vial containing sucrose solution. Small holes in the membrane allowed ants to feed through the membrane. Argentine ant feeding was enhanced at vials with the pheromone by 29% at four hours and 33% at 24 h when compared to control vials without pheromone. In a more recent study, (Z)-9-hexadecenal was added to a commercial thiamethoxam gel bait and ant response evaluated in the lab and field. In lab studies, more than 2-fold as many ants fed on the pheromone-treated bait and, as a result, ant mortality after seven days was significantly greater than in the standard commercial bait [23]. In field trials of 10 Argentine ant-infested homes, ≈2–3-fold more pheromone-assisted bait was consumed than bait not amended with the pheromone. By the fourth week post-treatment, the commercial bait and the commercial bait + pheromone had achieved a 42% and 74% reduction in ant activity compared to pre-treatment activity, respectively. For ethical reasons, this study did not have an untreated control. (Z)-9-hexadecenal might also aid spray treatments in an attract-and-kill approach. Small quantities of (Z)-9-hexadecenal were attractive to Argentine ants, and in field experiments pulled them short distances from existing trails and nests to areas treated with the pheromone, while in the laboratory Argentine ant mortality was enhanced when (Z)-9-hexadecenal was added to a common fipronil or bifenthrin spray [24]. The authors propose that adding (Z)-9-hexadecenal to common insecticidal sprays, termed pheromone-assisted technique, might reduce the size of the area treated while attracting and killing ants to the treated zone.

### 3.2. Use of Behavior-Modifying Chemicals

In the Formicidae, necrophoresis is the process by which dead individuals are removed from the nest and placed onto so-called “bone piles”—groups of dead ants typically moved away from the nest. Like many ant species, Argentine ants regularly engage in this behavior. A number of researchers have proposed that necrophoresis might help explain the activity of various contact insecticides, as healthy ants are contaminated and die as a result of handling insecticide-killed, and subsequently contaminated, corpses killed either by contact [25,26] or by being topically sprayed and killed [27]. In a study by Choe et al. [28], the chemistry underlying necrophoresis was explored. Previous work implicated decomposition products as the signal for necrophoresis [29]. In Choe’s study, it appears that the molecules that elicit the necrophoretic response are always present whether the ants are alive or dead [28]. The signal is masked by a combination of other highly volatile compounds, dolichodial and iridomyrmecin, preventing necrophoretic behavior toward living workers by their nestmates. The masking molecules are considerably volatile and so, when the ant dies and ceases production of the masking molecules, the signal molecules for necrophoresis are revealed, leading to faster response times than would be possible using decomposition products. Since these necrophoretic signal molecules are always present on live ants, it is possible to extract the compounds and elicit a similar necrophoretic response when they are placed on an inert object. Gochnour et al. [30], for instance, freeze-killed Argentine ant workers and then washed the dead ants in methylene chloride. Inert paper wicks were added to the extract and the solvent allowed to evaporate, leaving behind paper wicks coated with the residue of cuticular compounds that trigger necrophoresis. The wicks were then treated with formulated fipronil (*Termidor SC*, 9.1% fipronil; BASF Corporation, Research Triangle Park, NC, USA), an acute, contact insecticide, and offered to small groups of live ants. Live ants handled and removed the paper wicks at the same rate as freeze-killed, whole, intact nestmates; wicks treated with only the solvent (no cuticular compounds) were ignored by the worker ants. In treatments where wicks were treated with fipronil, mortality of ants was generally higher than untreated controls and only loosely dose related. Similarly, Wiltz et al. [31] coated inert corn cob granules with triolein or triolein + fipronil and offered the granules to foragers of small red imported fire ant, *Solenopsis invicta*, laboratory colonies. Triolein, thought to be a brood pheromone, is likely no more than a food source. Triolein enticed foraging ants to collect treated granules and move them into an artificial nest. Ants that collected granules coated with triolein + fipronil suffered 91% mortality, while ants provided granules coated with fipronil only suffered only approximately 47% mortality.

Buczkowski [32,33,34] and Buczkowski et al. [35] developed low-impact, novel delivery systems of an acute insecticide (fipronil) for the control of Asian needle ants, *Brachyponera chinensis* (Emery), Argentine ants, and black carpenter ants, *Camponotus pennsylvanicus* (De Geer). The techniques are target specific and do not impact non-target ant species. In their studies, insecticide-exposed, live termites served as ant “bait.” Termites were treated with fipronil (by contact or topical application) and, while still alive, placed into small laboratory ant colonies consisting of several hundred worker ants and some brood. As few as one fipronil-treated termite quickly killed 100% of ants in small colony propagules. When fipronil-treated termites were released in the field, ant activity was nearly eliminated after three to four weeks. The specificity of the bait (live ant prey) allowed for extremely effective control of the target ant species while reducing the amount of insecticide deposited into the environment. Buczkowski speculated that prey location was likely a combination of visual and chemical cues [32]. Indeed, much of ant (Formicidae) and Hymenopteran behavior is chemically mediated, and the future identification of prey-specific chemical cues that communicate prey, or food, might aid in the development of novel delivery methods of insecticides for pest ant control [30]. In the black carpenter ant study, 100% of 50 recipient ants were killed in three or four days when just one donor ant, exposed to 0.06% fipronil by topical spray or by being confined to a treated ceramic tile for one hour, was added to a group of 50 recipients [34]. When 20 of those recipients, killed by a single topically killed donor, became donors themselves and were mixed with 50 additional naïve ants, 90% of these tertiary level ants were killed after four days. In field studies, worker carpenter ants were collected from each of eight colonies, sprayed with 0.06% fipronil, and returned to their respective colony within one hour and their nestmates readily accepted them. Compared to common toxicants applied for ant control (e.g., pyrethroids), fipronil is slow acting. Ant activity in these “treated” colonies, as measured by the number of foragers counted on the bark of each tree (*C. pennsylvanicus* nests in hardwood trees), was reduced by 97% within a week and activity in all colonies (*n* = 8) had ceased after two weeks.

### 3.3. Mass Trapping

Newell and Barber attracted and then eradicated large numbers of Argentine ants, with “trap boxes,” from ant-infested citrus groves in California [36]. They built 2 by 2 by 3 ft. wood boxes, filled each with cotton seed and dead grass, and distributed them throughout a citrus grove. The top of each box was left open to facilitate water entry and decomposition of the vegetative matter. Within a month or so, many ants (workers, queens, and brood) began harboring in the boxes. To further facilitate ant movement into the traps, parts of the grove were cultivated by disturbing the soil and removing leaf litter. These sanitary practices helped induce a greater number of ants to enter the trap boxes. From Newell and Barber (p. 95, [36]), following their cultivation efforts: “In January 1911, the authors again visited this orchard and found all boxes filled almost to overflowing with enormous ant colonies.” Following attraction, they fumigated the ant-filled boxes and later noted that the number of ants in the grove had declined so dramatically that citrus trees were no longer troubled by honeydew producers and that tree health, and citrus production, was thereby restored. As temperatures begin to cool during the fall and winter, Argentine ants seek habitats protected from outdoor, ambient conditions. In northern California, for instance, Argentine ants readily enter homes during specific times of the year—primarily during cool, wet winters [37]. In their study, more than 60 homeowners in the San Francisco Bay area were queried weekly, for 78 consecutive weeks, and asked about invasions by Argentine ants that week. Ant invasions were most prevalent during the winter months when the weather was cold and wet. A similar phenomenon occurs in the Southeastern U.S. during the winter when ants move indoors during cold weather (Figure 1).

Argentine ants are particularly susceptible to desiccation because they exhibit high cuticular permeability [38]. As a result, they tend to be attracted to and harbor in microhabitats that exhibit relatively high humidity [39,40]. Driven by a somewhat similar concept, Silverman and Nsimba took advantage of the Argentine ant’s strong attraction to moist microhabitats and propensity to establish nest sites close to food resources by developing a soil-free method to collect large numbers of Argentine ants [41]. In their study, Petri dishes containing moistened Castone (a water absorbent substrate) were placed next to ant food sources inside a building, and Argentine ants from outside readily located into these dishes. Over time, they removed thousands of workers, a substantial quantity of brood, and numerous queens from the outdoor environment. In the absence of water sources, small cohorts of Argentine ant workers quickly moved into small Petri dishes containing moistened Castone [42,43], and Schilman et al. [38] showed that Argentine ants, when given a choice, were attracted to and harbored in small cups where the relative humidity was >90%. For decades, Argentine ant researchers have routinely taken advantage of the Argentine ant’s susceptibility to desiccation when separating recently collected ants from the field from their leaf litter debris. Ants, mixed with soil and leaf litter debris, are returned to the lab and lay exposed in open containers containing some form of enclosed, moistened microhabitat, typically one or more Petri dishes filled with plaster or Castone capable of slowly releasing water. As their leaf litter habitat dries, ants (workers, queens, and brood) predictably move into the debris-free, moistened containers. In a field study, Silverman et al. [44] attempted to take advantage of the propensity of Argentine ants to nest in moisture-retaining mulch by preparing trap-mulch to entice Argentine ants to nest, followed by spot applications of a non-repellent insecticide to the areas where the ants had concentrated, but the technique was marginally successful. Enzmann et al. [45] were able to manipulate the foraging pattern of Argentine ants based on the location of water sources. In their study, food (50% sugar water and scrambled eggs) was less important than water in manipulating the foraging patterns of Argentine ants in a hot, dry environment in southern California. They proposed that a combination of ornamental, xeric-adapted plants placed next to the structure combined with water resources placed away from the structure may lure ants away from structures that they might otherwise infest.

### 3.4. Hydrogels, “Virtual” Baiting, and Exceedingly-Low Bait Concentrations

Baits are target specific and take advantage of the eusocial nature of ants. Recently, there have been improvements to ant bait formulations [46,47,48,49]. Water-storing crystals, or hydrogels, are absorbent polymers used in situations where water conservation is important. When water is added to the dry granules they swell and can absorb hundreds of times their weight in water. When absorption is complete, the hydrogels take on a gelatinous appearance. Hydrogels, as tools in pest management, were recently reviewed [50].

Hydrogels dry quickly in dry environments, and dry baits are less preferred by foraging ants. Due to water loss, after four hours of aging ≈40% fewer ants fed on the gels, compared to the fresh gel and, as a result, gels aged for four hours or more were less palatable and less efficacious than fresh baits or baits aged for up to two hours; after 8 h of aging there were 65% fewer ants found feeding on the bait, resulting in only ≈55% worker mortality and ≈10% queen mortality in a no-choice offering to starved ants [46]. A more environmentally friendly alginate hydrogel, containing 0.0001% thiamethoxam (i.e., 1 mg thiamethoxam in a liter of 25% sugar water), was developed [49] and performed similarly to a 0.00015% polyacrylamide hydrogel [48]. In their studies, hydrogels experiencing ≥50% water loss were significantly less preferred by foraging Argentine ants than hydrogels experiencing ≤25% water loss. When held under dry conditions (0% or 32% humidity), the gels lost ≈70% to ≈90% water content in just 8 h; hydrogels held at 75% relative humidity and on wet sand had lost approximately ≈25% water after 8 h, while those held at 75% humidity but on a dry substrate lost ≈50% of their water [49]. Water loss was similar in Rust et al. [48], wherein water loss regression models were calculated. In their study, a polyacrylamide hydrogel containing 0.00015% thiamethoxam held at 0% and 33% humidity lost half of its water in 9–12 h, while the same gel held at 55% and 75% humidity did not lose 50% of its water until ≈1 and ≈2+ days, respectively. Hydrogel baits will likely perform better in environments with elevated, minimum daily humidity, warm temperatures, and perhaps more predictable and higher rainfall. In the U.S., this environment is defined by summer and fall in the Southeast, when Argentine ants are most problematic to property owners.

The 0.0001% [49] or 0.00015% [48] thiamethoxam hydrogels, offered to small, laboratory colonies of Argentine ants for 24 h in a no-choice situation, after having been starved for three days, killed 89% of workers and 100% of queens after three to five days. Interestingly, an experimental hydrogel containing 60% less thiamethoxam (i.e., 0.00004%) still killed 100% of queens and workers on days 7 and 14. Similarly, in a no-choice laboratory study, fresh polyacrylamide hydrogels containing 0.0007% thiamethoxam eliminated starved workers in approximately two days and queens in approximately five days [46,47]. In a more recent laboratory study, alginate hydrogels containing one of nine insecticides were evaluated against small colonies of Argentine ants (300 workers and two queens) in a no-choice study following three days of starvation [51]. Hydrogels containing dinotefuran, thiamethoxam, imidacloprid + fipronil, and spinosad were most effective.

Hydrogel baits have performed well in field studies. Hydrogels containing minimal amounts of thiamethoxam reduced Argentine populations in a plum orchard in South Africa [47], in citrus in California [52], on the environmentally sensitive Santa Cruz Island, California [53], and around structures in California [49]. After two bait applications (day 0 and week 4), a 0.0001% thiamethoxam alginate hydrogel bait reduced Argentine ant populations around five California houses by ≈42%, ≈68%, and ≈80% at 1, 4, and 8 weeks [49]. Just 10 mg of active ingredient was required to obtain such results among the baited homes. Several polyacrylamide hydrogels containing various thiamethoxam concentrations were applied twice (early August and late September) to Argentine ant-infested plots on Sant Cruz Island, CA, USA. Ant populations in baited plots had declined by 88% one week after the second bait application [48].

Choe et al. [54] developed a novel approach to controlling Argentine ants in a sensitive area where the eggs and chicks of an endangered bird are at risk of predation from foraging ants. The technique, termed virtual baiting, takes advantage of the Argentine ant’s preference for sweet liquids. Virtual baiting requires ants to recruit to a preferred food source, such as sugar water, and to access the food workers must crawl over a surface treated with a residual, contact insecticide. In their prototype station, residual fipronil was contained and inaccessible to non-target organisms and the degradative effects of sunlight but crawled on by worker ants as they recruited to the sugar water food source.

### 3.5. Food Source Reduction

A primary food source of warm-weather Argentine ant populations is honeydew produced by aphid and scale insects feeding on new and emerging growth of trees and perennial shrubs. Argentine ants exploit this predictable, carbohydrate-rich food source for much of the warm season. To facilitate direct and quick access to their food source, Argentine ants commonly nest on the ground and close their food source [41]. Commonly, this is at the base of trees, often in leaf litter or mulch, where they are foraging [55]. Rust et al. [2] proposed that Argentine ants might be controlled by eliminating Argentine ants’ primary food source, honeydew, by conducting foliar treatments targeted at reducing populations of honeydew-producing homopteran populations in vegetation. Several studies have addressed this approach. Brightwell and Silverman excluded Argentine ants from red maple trees infested with honeydew-producing terrapin scales by maintaining an 8 cm-wide band of Tanglefoot (a sticky substance) around the tree’s trunk [55]. Although Tanglefoot was not aimed at reducing scale numbers, it completely prevented ants from foraging into the tree’s canopy, eliminating the scale insect population probably from parasitism. Predictably, Argentine ant nests at the base of these trees were eliminated because access to nearby food (i.e., the tree into which they were foraging) was eliminated. The researchers had reasoned that adding a liquid ant bait to the system might replace the food source (honeydew) lost to Tanglefoot exclusion, thereby driving the ants to enhance their consumption of a liquid, boric acid bait. They surmised the quantity of bait was probably insufficient to draw ants to the bait in significant enough numbers to test the theory. In a follow-up study, this time with larger bait dispensers, conducted at the same location as [55], the same researchers treated scale-infested trees with a horticultural oil, an imidacloprid soil drench, and a dicrotophos tree injection [56]. Neither the oil nor the soil drench was effective at reducing scale numbers, and although dicrotophos injection was only partly effective (i.e., dicrotophos reduced the rate of scale population increase compared to trees not injected with dicrotophos), it still resulted in a reduction in (a) Argentine ant nests at the base of trees and (b) the number of ants foraging into the tree’s canopy, but it did not eliminate the scales as did Tanglefoot [55]. Unfortunately, the addition of a liquid bait did not enhance the effect of dicrotophos treatment even though access to bait was made greater than in their previous study [55]. The dicrotophos treatment did not drive ants to the liquid bait in sufficient numbers to see significant reductions in either the number of nests at the base of trees or the number of foragers moving into the tree’s canopy, likely because honeydew production was not eliminated from the system and was competing with the liquid bait. In the end, the technique has merit, and with larger bait dispensers, elimination of competitive honeydew, and perhaps a new active ingredient (e.g., a low dose of thiamethoxam), the approach is worthy of further pursuit. In a proof-of-concept laboratory study, Suiter (unpublished) demonstrated that Argentine ants abandoned foraging in aphid-infested pepper plants following soil treatment with 0.10% thiamethoxam (*Optigard Flex*, Syngenta Corporation, Greensboro, NC, USA) (Figure 2).

### 3.6. Deterrents

Plant-derived compounds (essential oils) have a long history of use as insecticides and repellents in pest control [57,58,59] and have recently garnered some attention as deterrents to Argentine ant nest site selection. Although studies have discovered the practical uses of natural products for arthropod control, much remains to be explored in their implementation as control agents for the Argentine ant, especially under field conditions [42,43,57]. Recent attention to essential oils is partly due to their exemption from registration by the EPA under the Minimum Risk Exemption regulations in 40 CFR 152.25(f)(1). According to Isman, this has caused an increase in the development and production of essential oil-based insecticides, fungicides, and herbicides for commercial and residential use [57]. Plant essential oils have been tested as repellents and insecticides against *L. humile*; however, there have only been a limited number of laboratory tests and no field studies to date [42,43]. Guerra et al. [60] conducted a laboratory study to determine nine essential oils’ effectiveness at killing the black carpenter ant. Extracts of basil, lemon, citronella, clove, eucalyptus, peppermint, rosemary, tea tree, and thyme were used in this study at a 10% concentration in acetone. A 0.03% bifenthrin suspension, an acetone only, and an untreated control were also assayed. The treatments were applied topically to the ants and mortality was recorded daily for seven days. At seven days post-treatment, none of the essential oil extracts achieved mortality comparable to the bifenthrin positive control. Citronella and tea tree extracts had significantly higher mortality than the acetone only and untreated controls, but did not exceed 33% mortality, compared to the 96% ant mortality in the bifenthrin treatment.

Wiltz et al. [43] evaluated the deterrent and toxic properties of six essential oils (tea tree, peppermint, lemon, citronella, and basil) to Argentine ants and red imported fire ants. In a choice test designed to evaluate deterrence, starved ants were given the choice to cross an essential oil-treated paper bridge or a solvent-treated (control) paper bridge to reach food and water. Concentrations ≥20 µL oil per cm^2^ were deterrent to both ant species; eucalyptus, however, was not deterrent to either ant species. When forcibly exposed to the extracts in a no-choice situation at 0.40 µL oil per cm^2^, citronella was the only treatment to kill 100% of Argentine ants after 24 h, while peppermint oil killed 89.8% of exposed ants and tea tree oil killed 85.7%. Citronella killed 50.6% of fire ants, but no other oil killed more than 6.2% of the fire ants. Mint oil granules were toxic and deterrent to red imported fire ants in laboratory studies and showed some promise as a mound treatment in the field [61]. When fire ants were forcibly exposed to mint oil granules in lab tests, both continuously and in limited exposures, mortality was dose and time related. Over a range of doses, the mint oil granules were deterrent to fire ants in a Petri dish bioassay where half the floor was treated with granules and the other half was granule-free. In field tests, the repellent nature of the granules caused mounds to relocate. In an extensive series of field trials, d-limonene and products containing d-limonene, were as effective or nearly as effective as a diazinon standard in eliminating ants’ activity in mounds [62]. D-limonene is an extract of citrus peel.

In a no-choice laboratory study designed to evaluate the harboring-deterrent potential of various essential oils to Argentine ants, Scocco et al. [42] applied one of three concentrations (10%, 1%, and 0.10%) of five extracts (clove, cinnamon, wintergreen, spearmint, and peppermint oils) to small, moistened ant harborages and assessed ant movement into the harborages when their choice was to remain exposed (no alternative harborage present). Argentine ants have a strong attraction to moistened microhabitats [39,40], and when oil was not present 82% of ants moved into the harborage after just 2 h. In this no-choice-of-harborage test all extracts at all concentrations deterred ant harboring when the extracts were fresh (2 h old). After seven days, the 10% and 1% extracts remained deterrent, while only spearmint retained its deterrent property at the 0.10% concentration. Meissner and Silverman published several articles on using cedar mulch, as an alternative to other mulch types, for the behavioral modification of nesting by Argentine ants [63,64]. Argentine ants commonly nest in mulch and leaf litter because of the soil moisture-holding ability of these substrates. In their studies, cedar mulch deterred Argentine ants from nesting, was toxic to Argentine ants in laboratory studies, and was more efficient than pine straw or cypress at deterring ants from nesting.

In laboratory studies, farnesol and methyl eugenol were deterrent to Argentine ant foraging [65]. Furthermore, in field studies, Shorey et al. [66,67] tested 19 chemicals and various delivery methods for their ability to keep Argentine ants from foraging into the canopy of lemon trees infested with honeydew producers. Farnesol was clearly a foraging deterrent, but so too were methyl eugenol, β-citronellol, and bornyl acetate.

### 3.7. RNA Interference (RNAi)

The use of RNAi as an urban pest ant management tool is in its infancy. In short, RNAi occurs when long double-stranded RNA (dsRNA) enters a cell, is sliced into smaller fragments which bind to proteins that use one of the strands as a guide to find a messenger RNA with a complementary sequence. This complementary messenger RNA is then degraded and thus unable to be translated into the intended protein [68]. In this way, the gene is silenced, and if the protein it encodes is vital for survival, the dsRNA results in insecticidal activity. Several researchers have injected dsRNA into ant pests and documented negative effects, but injecting RNA into insects is not viable for pest ant management. The two studies that have addressed potentially more viable options by introducing sugar-based RNAi diets to ants had limited results but provide proof of concept [69,70]. In his PhD dissertation work, Welzel introduced dsRNA of ADP, ATP carrier protein 2 in sugar water to groups of 10 Argentine ant workers [69]. After 6 days of continuous feeding, 52% of the workers were dead compared to 10% in the sugar-water control on day 7. Gene expression was reduced by 84% after 3 days of feeding. This is a first step to determine if the technology works in Argentine ants but falls short of demonstrating effects on an entire colony. Double-stranded RNAi of six housekeeping genes produced through bacterial gene expressions was incorporated into a glucose agar diet and fed to groups of 30 *Nylanderia fulva,* the tawny crazy ant, workers [70]. Reduced gene expression was noted for all six genes; however, survival was only significantly reduced by inhibition of two of these genes and then only by 15%. Should RNAi bait technology for ants be perfected, the benefits could resemble that of many of the alternate management practices described above—it could be target specific, reduce the reliance on chemical pesticides, and concerns about environmental contamination. As it stands, much improvement is needed to increase RNAi levels in the ants, for these levels to persist longer and be distributed to other members of the colony before this technique can be used as a successful ant management technology.

## Figures and Tables

**Figure 1 insects-12-00487-f001:**
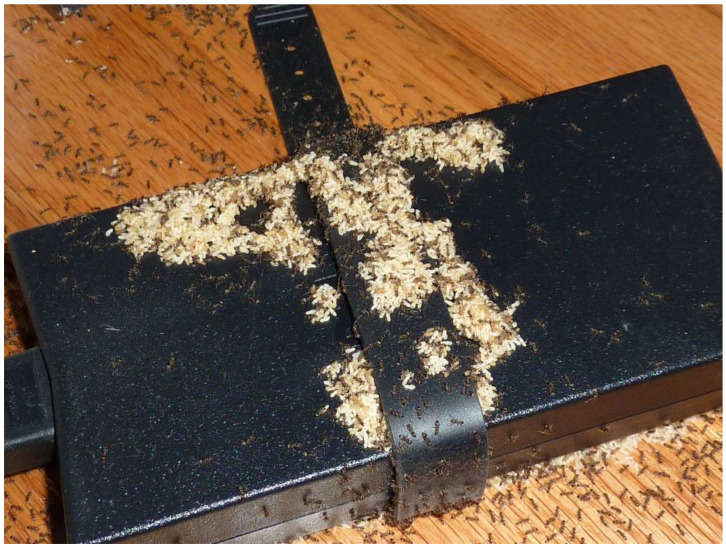
During cool winter weather in Georgia (USA) (October through March), Argentine ants routinely move indoors [37]. In this image, on a cool October evening in Georgia, Argentine ants located and recruited to a warm computer battery charger located indoors, where grams of brood were relocated (photo, D. Suiter).

**Figure 2 insects-12-00487-f002:**
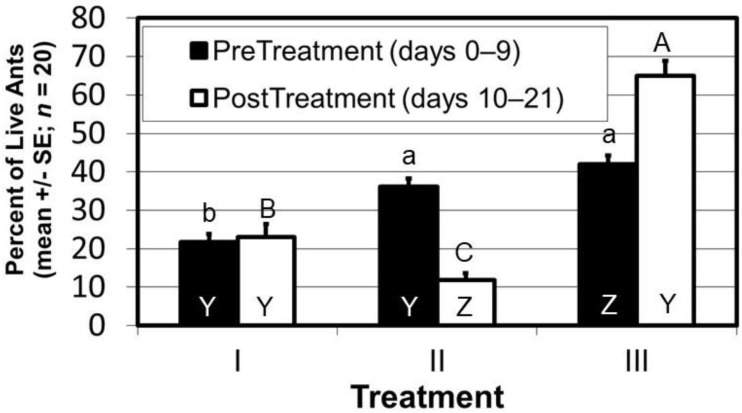
In a closed-system, Argentine ants were allowed to forage among three pepper plants and the number of ants on each plant counted daily for 21 days. Treatment I was a pepper plant never infested with aphids; Treatment II was a pepper plant infested with aphids from day one and on day nine the soil was injected with 75 mls of 0.10% *Optigard Flex* (thiamethoxam); Treatment III was a pepper plant infested with aphids for the entire 21-day study. In the nine days prior to thiamethoxam soil treatment ants foraged equally in pepper plants infested with aphids (Treatments II and III) and more than in the pepper plant that was not infested with aphids (Treatment I) (black bars, a–b, one-way ANOVA: F = 22.0; *p* < 0.0001; df = 2, 57). In the 11 days after thiamethoxam injection ants abandoned the thiamethoxam-treated plant (Treatment II) and increased foraging in the lone aphid-infested plant (Treatment III) (white bars, A–C, one-way ANOVA: F = 72.5; *p* < 0.0001; df = 2, 57). Ant Foraging Pre- versus Post-Thiamethoxam Injection: The percentage of ants foraging in the aphid free plant (Treatment I) did not differ pre- and post-treatment (black vs. white bars, Y-Y, *t*-test: *t* = 0.36; *p* = 0.7195; df = 38). After soil injection with thiamethoxam the percentage of ants foraging in the thiamethoxam-treated plant (Treatment II) dropped (black vs. white bars, Y-Z, *t*-test: *t* = −8.24; *p* < 0.0001; df = 38), while the percentage of ants foraging in the remaining aphid-infested plant (Treatment III) increased (black vs. white bars, Z-Y, *t*-test: *t* = 4.94; *p* < 0.0001; df = 38).

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
