# Peer review of "Alternative Methods of Ant (Hymenoptera: Formicidae) Control with Emphasis on the Argentine Ant, Linepithema humile"

_insects, 2021, doi:10.3390/insects12060487_

Round 1

Reviewer 1 Report

This review paper is well written. Information will be helpful to researchers and pest management professionals to understand the alternative ant management strategies. While the authors centered their discussion on Argentine ant, I hope they are more inclusive on the alternative methods adopted for managing urban ant pests. Environmental modification can be important to discourage ants from establishing colonies (for example: removing clutter around homes, ground cover type (mulch vs no mulch, or other types of ground cover) Homeowners may place ant infested flower pots in water to submerge ant nest and force ants coming out and kill them. Boiling water is used to kill nests around homes.

Line 54. It would be helpful to provide some data on the prevalence of different treatment methods and insecticide usage data for ant control.

Line 81. Why baiting is more labor intensive than spraying? I thought spreading bait is quite convenient compared to using a tank spray.

Line 212. Add city and state name.

Line 225. Add author “(Ge Geer)”.

Line 277. Is this a battery charger?

Figure 2. Font is too large. I suggest delete 1-6 explanations after Figure 2. Change x-axis label to I, II, III and explain them in the figure legend. Change x-axis title to Treatment.  What is the main purpose of this figure? I think the main purpose is to compare the percent of live ants on plants before and after treatment. The data show Optigard Flex treatment significantly reduced percentage of live ants on plants; whereas the other two treatments did not.  It does not make sense to compare the post-treatment percentage among the treatments (with symbols B, C, A) because the pre-treatment percentages are significantly different (with symbols b, a a).

Line 437. EPA

Line 444. Delete “(De Geer)”.

Author Response

  1. The work cited by Klotz et al. is an accurate assessment of typical ant control methods. Data on practitioner usage does not exist, to my knowledge. Klotz's excellent series of articles in Sociobiology reflects well current industry standards.
  2. Klotz et al have data on this, and as it turns out baiting is more labor intensive because of return visits when baiting. Initially, yes, baiting is less labor intensive, but in the long run baiting requires more time commitment. Fipronil changed this.
  3. Done
  4. Done
  5. Done
  6. All suggestions regarding Figure 2 have been implemented. These suggestions significantly improved the figure.
  7. Done
  8. Done

Reviewer 2 Report

This review article (insects-1202354) by Suiter et al. is well motivated, the structure is appropriate, and the manuscript is well written. The originality of this manuscript is that it goes beyond the simple exploration of literature, the description of events and the listing of different alternatives to chemical treatments for argentine ants (AA). It highlights many recent results and comments on their possible integration into IPM programs for AA. The only part that really needs to be reworked in my opinion is the section on hydrogel beads as some key information is missing from this section; my comments are below.

L339-345: More recent study has indicated that in addition to thiamethoxam, dinotefuran, fipronil/imidacloprid, and Spinosad hydrogels are also efficacious against AA. More specifically, Milosavljevic and Hoddle (2021; https://doi.org/10.1093/amt/tsab072) assessed the efficacy of nine different insecticides with previously demonstrated efficacy against AA (including thiamethoxam) when delivered to ants via alginate hydrogel beads, under lab conditions. In this study, all treatments reduced the number of AA workers and queens when compared with the untreated control on all posttreatment days and overall. However, the four abovementioned chemicals caused significantly higher mortality of workers and queens than other treatments or the 25% sucrose-only hydrogels. There were no significant differences in worker and queen mortalities between the low, medium, and high concentrations of thiamethoxam, dinotefuran, fipronil/imidacloprid, and Spinosad hydrogels on any posttreatment day. Because these insecticides have different modes of action, they may collectively mitigate resistance development when used together in a rotation program for AA control. Field trials are needed to evaluate efficacy of best performing insecticides and rates when delivered to AA using hydrogels. This information should be added to your revised article and the above mentioned reference should be cited in my opinion. 

Author Response

1. I obtained this most recent AMT article, read and digested it, and have incorporated it's results into the MS. I missed this. This article has improved the coverage of hydrogels.

Reviewer 3 Report

I found the review extremely interesting and well prepared. I think those interested in urban pest management will find this extremely useful. Just a few thoughts and suggestions.

Lines 20-21. I think the underlying goal has been the reduction of pesticides in urban water runoff. At least in CA, the focus has been on the Clean Water Act.

Lines 43-53. I find this an odd introduction. I would think that section 2 would better serve sa an introduction because the main focus of the review is on urban ant control.

Lines 92-129. There is a recent book by Goh et al. 2019. In pesticides in Surface Water, Monitoring, Modelling, Risk assessment, and management. ACS, Washington DC. Greenberg and Rust provide a review of nearly 10 years of research in pesticide runoff from homes treated for Argentine ants. This is probably a good reference to add.

Author Response

  1. Introduction deleted, and section 2 starts the beginning of the edited MS. Good suggestion.
  2. It took time, but I obtained Greenberg and Rust's chapter (23) on urban issues and ant control, read it, and cited and included it in the revised MS.